# A Novel Carotenoid with a Unique 2,6-Cyclo-ψ-End Group, Roretziaxanthin, from the Sea Squirt *Halocynthia roretzi*

**DOI:** 10.3390/md20120732

**Published:** 2022-11-24

**Authors:** Takashi Maoka, Chisato Tode

**Affiliations:** 1Research Institute for Production Development, Shimogamo-Morimotocho, Sakyo-ku, Kyoto 606-0805, Japan; 2Instrumental Analysis Center, Kobe Pharmaceutical University, Motoyamakita-Machi, Higashinada-ku, Kobe 658-8558, Japan

**Keywords:** carotenoid, 2,6-cyclo-ψ-end group, roretziaxanthin, sea squirt, *Halocynthia roretzi*

## Abstract

A novel carotenoid with a unique 2,6-cyclo-ψ-end group, named roretziaxanthin (**1**), was isolated from the sea squirt *Halocynthia roretzi* as a minor carotenoid along with (3*S*,3′*S*)-astaxanthin, alloxanthin, halocynthiaxanthin, mytiloxanthin, mytiloxanthinone, etc. This structure was determined to be 3-hydroxy-1′,16′-didehydro-1′,2′-dihydro-2′,6′-cyclo-β,ψ-carotene-4,4′-dione by UV–VIS, MS, and NMR spectral data. The formation mechanism of roretziaxanthin in the sea squirt was discussed.

## 1. Introduction

Tunicates are marine invertebrates belonging to Protochordata that can show bright red colors due to containing various carotenoids such as (3*S*,3′*S*)-astaxanthin, alloxanthin, mytiloxanthin, mytiloxanthinone, halocynthiaxanthin, amarouciaxanthin A, and amarouciaxanthin B [1,2,3,4]. The sea squirt *Halocynthia roretzi* is an edible tunicate that is classified in the family Pyuridae in Protochordata, and has been consumed as sashimi, vinegared food, and salted food in East Asia, especially in South Korea and the northeast region of Japan. (3*S*,3′*S*)-astaxanthin, alloxanthin, diatoxanthin, zeaxanthin, diadinochrome, fucoxanthinol, halocynthiaxanthin, mytiloxanthin, and mytiloxanthinone have been reported as the principal carotenoids in *H. roretzi* (Figure 1) [3]. In the present study, we isolated another new carotenoid, having a unique 2,6-cyclo-ψ-end group, named roretziaxanthin (**1**) from *H. roretzi* as a minor carotenoid. This manuscript reports the isolation and structural elucidation of this new carotenoid.

## 2. Results and Discussion

A new crimson-colored carotenoid (**1**) named roretziaxanthin (1.5 mg, 1% of the total carotenoids) was isolated from the tunic (4 kg) of the sea squirt *H. roretzi*. This carotenoid (**1**) showed visible light absorption maxima at (463), 482, and 505 nm in diethyl ether (Et_2_O) (Appendix A). The molecular formula of **1** was determined to be C_40_H_50_O_3_ by high-resolution FAB MS and ESI MS data (Appendix A). The ^13^C-NMR spectrum, including DEPT experiment, showed forty carbon signals: nine CH_3_, two sp^3^ CH_2_, one sp^2^ CH_2_, two CH, fourteen alkenes CH, and twelve quaternary carbons. ^13^C-NMR signals of this carotenoid at δ 200.42 (C4) and 207.79 (C4′) indicated the presence of two carbonyl groups in the molecule. Similarly, the ^13^C-NMR signal at δ 69.19 (C3) and ^1^H-NMR signals at δ 4.32 (H3) and 3.69 (OH) showed the presence of a hydroxy group in **1** (Appendix A). ^1^H- and ^13^C-NMR signals of **1** were assigned by COSY, NOESY, HSQC, and HMBC experiments (Table 1, Appendix A). From the ^1^H- and ^13^C-NMR data, this carotenoid consisted of two end groups and a polyene chain with nine conjugated double-bond systems. The presence of a 3-hydroxy-4-keto-β-end group (C1 to C6, C16, C17, C18) in **1** was revealed by a comparison of ^1^H- and ^13^C-NMR data with those of astaxanthin [5]. The structure of the polyene chain with all *E* geometry was also confirmed by ^1^H- and ^13^C-NMR data and NOE correlations [5]. The remaining structural part consisted of two methyl groups (C17′, C18′), one sp^3^ methylene group (C3′), one sp^2^ methylene group (C16′), one sp^3^ methin group (C2′), and four quaternary carbons (C1′, C4′, C5′, C6′), including one carbonyl group (C4′, δ 207.79). The structure of this end group was elucidated by detailed analysis of HMBC correlations, as shown in Figure 2. The HMBC correlations from H2′ to C3′ and C6′, from H3′ to C2′ and C4′, and from H18′ to C4′, C5′, and C6′ revealed the existence of a cyclopentene-4′-one ring attached to a methyl group (C18′) at C5′. Furthermore, HMBC correlations from H2′ to C16′and C17′, from H16′ to C2′and C17′, and from H17′ to C2′and C16′ indicated that the propylene group (C1′, C16′, C17′) was attached at the C2′ position of this cyclopentene-4′-one ring. Moreover, HMBC correlations from H7′ to C2′and C6′, and from H8′ to C6′ showed that this end group was attached at the C7′ position of the polyene chain, as shown in Figure 2. This novel end group could not be described from the ordinal IUPAC semi-systematic nomenclature of carotenoids [6]. However, according to the new nomenclature rule proposed for 2,6-cyclo-lycopene derivatives of carotenoids by Khachik et al. [7,8,9], this novel end group was regarded as a derivative of the 2′,6′-cyclo-ψ-end group and designated as 4′-keto-1′,16′-didehydro-1′,2′-dihydro-2′,6′-cyclo-ψ-end group. The NOE correlations between H8′ and H2′, between H8′ and H16′, and between H8′ and H17′ indicated that the propylene group in this end group was located on the same side of the polyene chain, as shown Figure 2. Furthermore, the product ion of FAB MS/MS at *m/z* 425 (M-153, C_9_H_11_O) was in agreement with the elimination of this end group from the molecule due to the cleavage of a single bond between C6′ and C7′. Therefore, the structure of roretziaxanthin (**1**) was determined as 3-hydroxy-1′,16′-didehydro-1′,2′-dihydro-2′,6′-cyclo-β,ψ-carotene-4,4′-dione (Figure 2). This carotenoid had two asymmetric carbons at C3 and C2′. However, the chiralities of these asymmetric carbons could not be determined by CD spectral data. Astaxanthin presented in sea squirts only shows (3*S*,3′*S*) chirality [3]; therefore, (3*S*) chirality was proposed for roretziaxanthin (**1**) (Appendix A).

In general, animals do not synthesize carotenoids de novo, and so those found in animals are either directly obtained from food or partly modified through metabolic reactions. Sea squirts are filter feeders. They feed on micro-algae such as diatoms, dinoflagellates, blue–green algae and green algae, and obtain carotenoids from these dietary sources. The major carotenoid in diatoms is fucoxanthin. Fucoxanthin is metabolized to fucoxanthinol, halocynthiaxanthin, mytiloxanthin, and mytiloxanthinone in sea squirts [1,2,3,4]. (3*S*,3′*S*)-astaxanthin is an oxidative metabolite of zeaxanthin ingested from blue–green algae [1,2,3,4]. Alloxanthin, diatoxanthin, and diadinochrome is also accumulated from dietary algae. Roretziaxanthin (**1**) has a unique 2,6-cyclo-ψ-end group in its molecule. Carotenoids with a 2,6-cyclo-ψ-end group have not been found in marine animals or algae. Only a few kinds of carotenoids having a 2,6-cyclo-ψ-end group, such as 2,6-cyclolycopene-1,5-epoxide, 2,6-cyclolycopene-1,5-diol, 1,16-didehydro-2,6-cyclo-lycopene-5-ol, and 2′,6′-cyclo-γ-carotene-1′,5′-diol, have been isolated from the tomato, gac (*Momordica cochinch* inensis), and human serum, as oxidative metabolites of lycopene or γ-carotene [7,8,9,10,11,12,13]. Khachike et al. [7,8,9] and Lu et al. [10] demonstrated that the oxidation of lycopene with *m*-chloroperbenzoic acid or hydrogen peroxide yielded 2,6-cyclo-lycopene derivatives, having a 2,6-cyclo-ψ-end group. From the results of these studies, roretziaxanthin (**1**) might be formed from rubixanthin (β,ψ-caroten-3-ol), as shown in Figure 3. However, rubixanthin has not been reported in sea squirts or marine algae [6]. On the other hand, rubixanthin was found in some species of Flavobacterium [6,14,15]. It has been reported that some carotenoids in marine invertebrates are produced by symbionts [1,2,16]. Therefore, roretziaxanthin (**1**) might be produced by symbionts in sea squirts.

## 3. Materials and Methods

### 3.1. General

The UV–visible (UV–VIS) spectra were recorded with a Hitachi U-2001 (Hitachi High-Technologies Corporation, Tokyo, Japan) in diethyl ether (Et_2_O). The positive ion fast atom bombardment mass spectrometry (FAB MS) spectra were recorded using a JEOL JMS-HX 110A mass spectrometer (JEOL, Tokyo, Japan) with *m*-nitrobenzyl alcohol as a matrix. The positive ion electro spray ionization time of flight mass (ESI-TOF MS) spectra were recorded using a Waters Xevo G2S Q TOF mass spectrometer (Waters Corporation, Milford, CT, USA). The ^1^H-NMR (500 MHz) and ^13^C-NMR (500 MHz) spectra were measured with a Varian UNITY INOVA 500 spectrometer (Agilent Technologies, Santa Clara, CA, USA) in CDCl_3_. The chemical sifts are expressed in ppm relative to tetramethyl silane (TMS) (δ = 0) as an internal standard for ^1^H-NMR and CDCl_3_ (δ = 77) as an internal standard for ^13^C-NMR. *J* values are given in Hz. The CD spectra were recorded in EPA [Et_2_O–isopentane–ethanol (5:5:2)] at room temperature with a Jasco J-500C spectropolarimeter (Jasco corporation, Tokyo, Japan). Preparative high-performance liquid chromatography (HPLC) was performed on a Shimadzu LC-6AD with a Shimadzu SPD-6AV spectrophotometer (Shimadzu Corporation, Kyoto, Japan) set at 450 nm. The column used was a 250mmX10mm i.d. 10 μm Cosmosil 5SL-II and 5C18 II (Nacalai tesque, Kyoto, Japan).

### 3.2. Animal Material

The sea squirt *H. roretzi,* cultivated at Yamada Bay in Iwate Prefecture, was purchased at a local fish market in Kyoto in October.

### 3.3. Isolation of Roretziaxanthin from the Sea Squirt H. roretzi

The tunic (4 kg) of the sea squirt *H. roretzi* was extracted with acetone (Me_2_CO) at room temperature. The Me_2_CO extract was partitioned with hexane/Et_2_O (1:1) and water. The hexane/Et_2_O layer was evaporated to dryness and chromatographed on silica gel using increasing Et_2_O in hexane. The fraction eluted with hexane/Et_2_O (2:8) was subjected to HPLC on silica gel with Me_2_CO/hexane (3:7), with a retention time (RT) of 14.1 min. Further purification to remove lipid impurities from **1** was performed by HPLC on ODS with methanol. Finally, 1.5 mg (1% of the total carotenoids) of roretziaxanthin was obtained as a crimson-colored powder.

### 3.4. Carotenoids from the Sea Squirt H. roretzi

The following carotenoids were isolated from the sea squirt *H. roretzi* along with roretziaxanthin. The fraction eluted with hexane/Et_2_O (2:8) was subjected to HPLC on silica gel with Me_2_CO/hexane (3:7), with an RT of 12.2 min to yield (3*S*,3′*S*)-astaxanthin (3.5 mg), and an RT of 15.0 min for mytiloxanthinone (6.5 mg) (Appendix A). The fraction eluted with Et_2_O/Me_2_CO (1:1) was subjected to HPLC on silica gel with Me_2_CO/hexane (3:7): RT of 15.0 min, zeaxanthin (3 mg); RT of 16.2 min, diatoxanthin (9 mg); RT of 18.0 min, alloxanthin (33 mg). The fraction eluted with Me_2_CO was subjected to HPLC on silica gel with Me_2_CO/hexane (3:7): RT of 19.0 min, diadinochrome (2.5 mg); RT of 21.2 min, mytiloxanthin (3.5 mg) (Appendix A); RT of 23.0 min, halocynthiaxanthin (3 mg) (Appendix A); and RT of 25.0 min, fucoxanthinol (2 mg). The identification of these carotenoids was described in a previous paper [3].

### 3.5. Spectral Data of Roretziaxanthin (**1**)

UV–VIS (Et_2_O): 463 (shoulder), 482, and 505 nm. HR-FAB MS: *m/z* 578.3756 (M^+^) C_40_H_50_O_3_ calcd for 578.3760. Product ions of FAB MS/MS (precursor ion M^+^): *m/z* 560 (M-H_2_O), *m/z* 486 (M-92), *m/z* 425 (M-153). HR-ESI MS *m/z* 601.3641 (M + Na^+^) C_40_H_50_O_3_Na calcd for 601.3658, *m/z* 579.3820 (M + H^+^) C_40_H_51_O_3_ calcd for 579.3838. Product ions of FAB MS/MS (precursor ion M + H^+^): *m/z* 561 (M + H-H_2_O), *m/z* 486 (M + H-92), *m/z* 473 (M + H-106), *m/z* 425 (M–153). CD (in EPA at room temperature): nm (∆ε) 225 (−4), 265 (−12), 300 (0), 325 (−1), 350 (0). ^1^H-NMR (500 MHz) and ^13^C-NMR (125 MHz) in CDCl_3_ are compiled in Table 1. The spectrums please see the Appendix A.

## 4. Conclusions

The structure of the new carotenoid roretziaxanthin (**1**), isolated from the sea squirt *Halocynthia roretzi*, was determined to be 3-hydroxy-1′,16′-didehydro-1′,2′-dihydro-2′,6′-cyclo-β,ψ-carotene-4,4′-dione by UV–VIS, MS, and NMR spectral data. Roretziaxanthin (**1**) has a unique 2,6-cyclo-ψ-end group in its molecule. Carotenoids with a 2,6-cyclo-ψ-end group have not been found in marine animals or algae. Roretziaxanthin (**1**) might be formed from rubixanthin through an epoxidation, cyclization, dehydration, or oxidation process, as shown in Figure 3 by symbionts in sea squirts.

## Figures and Tables

**Figure 1 marinedrugs-20-00732-f001:**
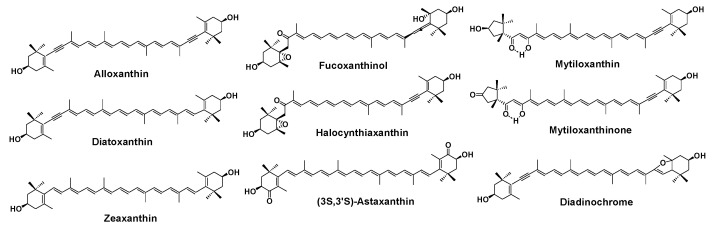
Principal carotenoids in sea squirt *H. roretzi*.

**Figure 2 marinedrugs-20-00732-f002:**
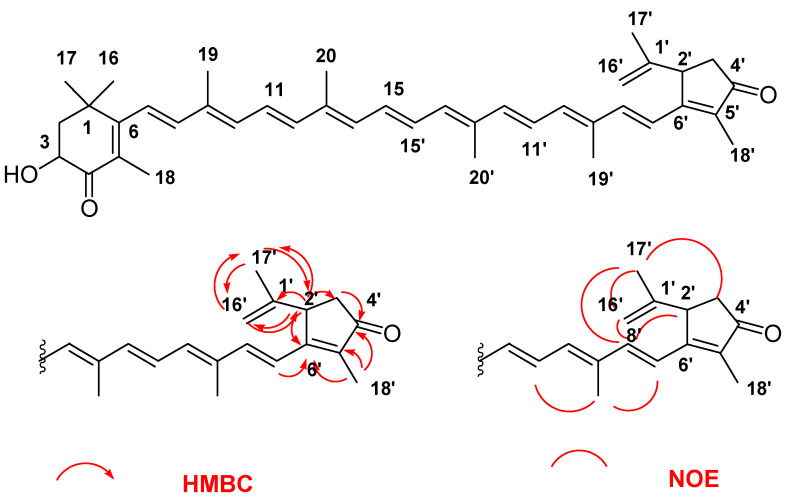
Structure of roretziaxanthin (**1**) and key HMBC and NOE correlations of 4′-keto-1′,16′-didehydro-1′,2′-dihydro-2′,6′-cyclo-ψ-end group of (**1**).

**Figure 3 marinedrugs-20-00732-f003:**
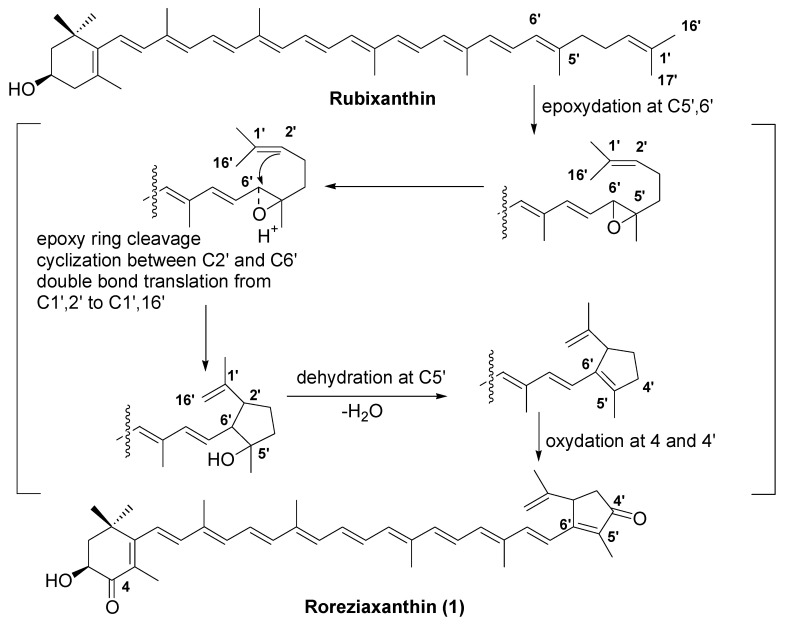
Possible formation mechanism of roretziaxanthin (**1**) from rubixanthin.

**Table 1 marinedrugs-20-00732-t001:** ^13^C-NMR (125 MHz) and ^1^H-NMR (500 MHz) data (CDCl_3_) of roretziaxanthin (**1**).

Position	^13^C-NMR	^1^H-NMR		Position	^13^C-NMR	^1^H-NMR	
	δ	δ	Mult.J (Hz)		δ	δ	Mult.J (Hz)
1	36.80		dd (13.5, 13.5)	1’	146.0		
2	45.38	1.81	dd (13.5, 5.5)	2’	46.23	3.78	d (7.5)
		2.15	ddd (13.5, 5.5, 2.0)				
3	69.19	4.32		3’	41.46	2.20	dd (18.5, 2.0)
						2.71	dd (18.5, 7.5)
4	200.42			4’	207.79		
5	126.78			5’	136.92		
6	162.25			6’	164.60		
7	123.33	6.22	d (16.0)	7’	120.59	6.62	d (15.5)
8	142.35	6.43	d (16.0)	8’	141.39	6.95	d (15.5)
9	134.66			9’	135.25		
10	135.16	6.31	d (11.0)	10’	137.51	6.37	d (11.5)
11	124.73	6.69~6.63	overlapped	11’	124.73	6.69~6.63	overlapped
12	139.67	6.45	d (15.5)	12’	140.49	6.49	d (11.5)
13	136.78			13’	136.92		
14	133.81	6.33~6.31	overlapped	14’	134.34	6.33~6.31	overlapped
15	130.65	6.69~6.63	overlapped	15’	130.95	6.69~6.63	overlapped
16	29.15	1.32	s	16’	112.90	4.88	t (2.0)
						4.95	s
17	30.75	1.21	s	17’	18.08	1.52	s
18	14.04	1.95	s	18’	8.40	1.88	d (2.0)
19	12.85	2.01	s	19’	2.48/12.81	2.00/2.01	s
20	12.59	1.99	s	20’	2.48/12.81	2.00/2.01	s
OH		3.69	d (2.0)				

Multiplicity: s: singlet, d: doublet, q: quartet, m: multiplet. ^1^H- and ^13^C-NMR signals of 19′ and 20′ are indistinguishable.

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
