# Peer review of "A Novel Carotenoid with a Unique 2,6-Cyclo-ψ-End Group, Roretziaxanthin, from the Sea Squirt Halocynthia roretzi"

_marinedrugs, 2022, doi:10.3390/md20120732_

Round 1

Reviewer 1 Report

This manuscript describes the structure of a new carotenoid. The carotenoid possesses a five-membered ring in the one terminal part. In spite of a small modification, this is an unusual structure. I think that  the configuration of C-2' in this molecule should be assigned. Why could not be determined?  ECD comparison may be solved for this determination.

Besides some points should be considered

1) compound number, e.g, 1 is bold type  1.

2)In the table 1, the 1H and 13C chemical shifts for 19' and 20' are        indistinguishable? Add the comment

3) In the table 1, usually the chemical shifts for 13C present to one decimal place and the coupling constant between H and H is also one decimal place. e.g. 2 to 2.0

4) In the NOE, the NOE correlation between H-2' and H-16' is observed.  Based on this correlation, discuss the configuration of C-2'

5) Please improve the chemical structure of 1 in the Figure 2. 

6) How much is the weight of the purified compound 1 which impurities are removed ?  

Author Response

Thank you reviewing our manuscript and polite and beneficial comments. We revised manuscript according to review comment.

About determination of absolute configuration of at C-2'. Determination of absolute configuration by CD or ECD needs for model compounds, which absolute configuration have been determined. In the case of roretziaxanthin, there are no model compounds to compare with CD or ECD. Furthermore, roretziaxanthin has been decomposed. Therefore, absolute configuration of roretziaxanthin cannot be determined now.

1) Compound numbers are written in bold,

2) In Table 1. 1H and 13C-BNR of 19' and 20' are in distinguishable. This point is described in footnote of Table 1. 

3) Table 1 is revised according to suggestion.

4) It is difficult to determine absolute configuration at C-2' from NOE correlation.

5) It is difficult to more improve Figure 2.

6) 1.5 mg is final obtained amount of roretziaxanthin. This point is revised L 152.

Reviewer 2 Report

comments in review file

Author Response

Thank you reviewing our manuscript and polite and beneficial comments. We revised typographical errors according to comments.

Line 99 an oxidative metabolite     This sentence describes several oxidative metabolites of lycopene and gamma-carotene. Therefore, we describe as plural form this sentence. 

Other revising points are revised according to suggestion. 

Reviewer 3 Report

attached

Author Response

Thank you reviewing our manuscript and polite and beneficial comments. We revised manuscript according to review comments.

High resolution mass spectral data of M+, M+H+. M+Na+ data of roretziaxanthin were still described in 1.5 Spectral data of roretziaxanthin (1) L 166~L169.  

Table 1 are revised according to your suggestion.

About formation mechanism of roretziaxanthin

Cleavage of the epoxy ring and formation of a new bond between C2' and C6' in Figure 3 is based Khachik's paper.  If only the stability of the carbocation formed after the cleavage of the epoxy ring is considered, a carbocation should be generated at C5', Then it must be attached the double bond of C1', 2' to form a four-membered ring. However, NMR spectra suggested that this compound have a five-membered ring. Therefore, we proposed the pathway of Figure 3. According to your suggestion, Figure 3 is revised.

Othe revising points are revised according to comments.

Reviewer 4 Report

This classic isolation/identification paper describes a new carotenoid and even an unknown end-group proving there is still new carotenoids under the sun to discover. The quality of the paper is high as one would expect from prof. Maoka. The discussion about possible biosynthetic ways is sound, based on all data they may have found the best hypothesis. I recommend to accept the paper after minor revision.

My remarks, amendments are the following:

Table 2.: in the header "Position". If possible move it right behind Figure 2.

line 143:  it is said here that other minor carotenoids were previously identified, however the cited article is from 1984 and eg. no 13C-NMR presented there. The authors should cite other articles where more extensive data can be found about these carotenoids, such as diadinochrome (2.5mg), fucoxanthinol (2mg), halocynthiaxanthin (3mg), mytiloxanthin (3.5mg), mytiloxanthinone (6.5mg). Maybe the authors themselves have eg. the 13C-NMR of these carotenoids. These data are important for carotenoid chemists so if they have not been published yet, they should be.

Chapter 3.3: In the description of column chromatographies please add which carotenoid elutes and in which order. Did the hexane:ether 2:8 fraction contain  only Roreziaxanthin, if not, what else?

Supplementary: please magnify the NMR spectra (and put them in landscape view) for better visibility. 

Author Response

Thank you reviewing our manuscript and polite and beneficial comments.

It is difficult to move header position in Table 2.

1H and 13C-NMR  data of mytiloxanthin, mytiloxanthinone and halocynthiaxanthin obtained in our laboratory are described in supporting materials as Tables.

Isolation procedure of another known carotenoids are described in Chapter 3.3,

Round 2

Reviewer 1 Report

My suggestion for determining the absolute configuration at C-2' is to compare the experimental CD with the calculated ECD spectrum. So authors need to calculate the ECD spectrum for roretziaxxanthin. The compound is not necessary to calculate the ECD spectrum

Author Response

Thank you provide beneficial suggestion. However, we are not familiar to calculation chemistry and have not experienced calculation of CD or ECD spectrum. Furthermore, we cannot access calculation program of ECD spectrum now.  Therefore, we cannot determine the absolute configuration of roretziaxanthin by calculation ECD spectrum now.  We hope to accept this manuscript for publication of this journal in the present form. We will try to determine the absolute configuration of this carotenoid with collaboration of calculate chemists.
